# Frame Registration for Motion Compensation in Imaging Photoplethysmography

**DOI:** 10.3390/s18124340

**Published:** 2018-12-08

**Authors:** Dmitry Iakovlev, Sijung Hu, Vincent Dwyer

**Affiliations:** Wolfson School of Mechanical, Electrical and Manufacturing Engineering, Loughborough University, Loughborough LE11 3TU, UK; D.Iakovlev@lboro.ac.uk (D.I.); V.M.Dwyer@lboro.ac.uk (V.D.)

**Keywords:** frame registration, imaging photoplethysmography (iPPG), remote sensing, heart rate measurement, motion artifact, motion compensation, signal processing, region of interest (ROI), healthcare

## Abstract

Imaging photoplethysmography (iPPG) is an emerging technology used to assess microcirculation and cardiovascular signs by collecting backscattered light from illuminated tissue using optical imaging sensors. An engineering approach is used to evaluate whether a silicone cast of a human palm might be effectively utilized to predict the results of image registration schemes for motion compensation prior to their application on live human tissue. This allows us to establish a performance baseline for each of the algorithms and to isolate performance and noise fluctuations due to the induced motion from the temporally changing physiological signs. A multi-stage evaluation model is developed to qualitatively assess the influence of the region of interest (ROI), system resolution and distance, reference frame selection, and signal normalization on extracted iPPG waveforms from live tissue. We conclude that the application of image registration is able to deliver up to 75% signal-to-noise (SNR) improvement (4.75 to 8.34) over an uncompensated iPPG signal by employing an intensity-based algorithm with a moving reference frame.

## 1. Introduction

The ability to monitor subcutaneous blood flow in living tissues has been an area of intensive biomedical research for decades [1,2,3]. Until recently laser Doppler and laser speckle imaging were the only viable non-contact techniques for mapping capillary blood flow [4]. Imaging photoplethysmography (iPPG) has since been used to demonstrate the feasibility of remote blood perfusion imaging as an inexpensive alternative method where tissue surface is illuminated by ambient [5] or artificial light [6,7,8], and modulated backscattered light is captured by an image sensor, typically a digital camera. However, the extracted iPPG waveforms have been found to be sensitive to optical distortions, in particular body motion, which is able to instantly change the amount of backscattered and stray light hitting the camera sensor [9,10]. Pioneering work on iPPG systems often still operates under the condition of a motionless subject [11,12] or at least very limited natural motion [13] during video recording, which tempers its acceptance and wide uptake in real-time clinical applications.

The main disadvantage of motion-uncompensated perfusion mapping is the loss of spatial resolution caused by the need to average over a relatively large block of pixels (varying with the application and the severity of motion) in order to overcome a poor signal-to-noise ratio (SNR). Focusing on the same skin area during the whole cardiac cycle is paramount in obtaining a high-resolution perfusion image of a rich capillary bed in body areas such as the face, palms, and fingertips, as otherwise neighboring regions would overlap in a single map and create undesirable blurring. Therefore, motion compensation becomes a crucial pre-processing step.

Progress in understanding subject motion and its compensation has been reported by applying a chrominance-based approach to color camera video frames [14,15]. Independent and principal component analyses (ICA and PCA) have been utilized to retrieve a motion-robust signal by returning a linear combination of the three color channels, assuming that the unknown cardiac-related pulsatile signal is *a priori* periodic, while the motion-related distortion is non-periodic [13,16]. A simple yet effective method of motion suppression is through object tracking by image registration [17], which has shown its usefulness as a pre-processing stage before more advance signal computation. Image registration is the process of aligning two or more images (usually called a template and targets) using a mathematical transformation in such a way that mutually registered images include overlapping scenes of the same features. Typical applications include remote environmental monitoring, motion stabilization in video cameras, multi-modal imaging of internal organs and tumors, and quality control on production lines. The rationale for utilizing image registration is to eliminate, or minimize, in-image motion of the tissue under examination, in advance of extracting PPG signals and constructing perfusion maps. However, studies systematically investigating the application of optical image stabilization by frame registration and its impact on iPPG signal quality have been somewhat neglected in the peer-review literature.

One motivation for this study, amongst others, is the lack of a standardized framework for assessing motion compensation schemes due to the variability of parameters, such as optical components selection, sensor resolution, distance to the target surface, field of view and an object’s motion amplitude, to name a few. Contemporary studies usually focus on discussing only some of these aspects, while each of them presents a unique challenge, and contribution, to the recovered iPPG waveform and derived vital signs. Therefore, the influence of each of these factors on the overall iPPG signal quality is difficult to quantify. Moreover, the fact that the iPPG peak–peak amplitude contributes as little as 0.6% of the total extracted signal [18] minimizes the ability to identify whether registered noise is caused by signal processing artifacts, variation in light uniformity, or the object’s motion vector with respect to a light source. Typical utilization of adaptive filters to isolate these noise components in the frequency domain also becomes impossible when the acquired motion is in-band with the cardiac activity.

The aim of this study is to understand how various parameters influence iPPG signal quality, starting from an object that could mimic the shape of human subject, but does not exhibit any time-varying pulsatile fluctuations associated with cardiac and respiratory functions. During the first stage, a high accuracy replica of a human palm is cast, and various noise sources are measured to establish a signal baseline when the object is static. In the second stage, low amplitude motion is modeled by exposing the prosthetic palm to a controlled displacement *w.r.t.* the optical system, by placing the cast on a movable platform. Four image registration algorithms are then applied to the video sets collected to assess their contribution to motion suppression in extracted spatially averaged signals. The final stage is dedicated to carrying over knowledge obtained throughout the previous steps and confirming preliminary findings with controlled live tissue experiments. Finally, a conclusion on the systematic approach and suitability of the optical image registration as a pre-processing step in iPPG waveform extraction is drawn.

## 2. Materials and Methods

### 2.1. Palm Mold Preparation

In order to compare various approaches to optical motion compensation, there needs to be a framework invariant to changes other than motion-induced ones during the experiment. It is anticipated that, if live skin tissue is used for image recording and the simulation of relative motion, it would be hard to identify the exact cause and to justify whether any changes in the morphology and quality of the extracted iPPG signal are to be attributed to the natural fluctuation in the vital signs over a short period, to the signal noise due to the variations in diffused backscattered and specular light, or to the quality of the motion registration algorithms.

Therefore, a new approach is proposed by creating a human palm model and replicating its physical attributes, such as its shape, curvature, protruding superficial blood vessels, and features such as wrinkles and birthmarks. Since the artificial model does not exhibit any cardiac pulse-related light backscattering and reflection, the effects of motion compensation on the extracted signal can be more closely evaluated.

The palm model was produced by lifecasting, where a three-dimensional copy was created replicating very small details including wrinkles, fingerprints, and pores with a high level of detail (Figure 1). The lifecasting process was approved by the Ethics Committee at Loughborough University, UK, and a participating subject signed a consent form prior to the procedure.

There are variations in the lifecasting process performed, depending on the body part being replicated, the level of detail required, and the reusability of the mold. The outline of the process used here is described below:*Model preparation:* A thin layer of hypoallergenic release agent was applied to an untreated skin surface to facilitate palm release and minimize adhesion to skin and hair.*Mold application:* To achieve a high level of detail, a mixture of non-toxic hypoallergenic alginate (Polycraft chromatic alginate, MB Fibreglass, UK) and water was poured into a tall plastic container followed by the slow insertion of the palm to avoid trapping air bubbles.*Casting:* High strength and tear resistant silicone was used as a casting material (Polycraft RTV condensation cure silicone rubber, MB Fibreglass, UK). A pigment (Polycraft dark flesh silicone pigment, MB Fibreglass, UK) was added to mimic Groups II and III of the skin classification system developed by Fitzpatrick [19]. The temperature was controlled to 20–24${}^{\circ }$ to avoid premature setting.*Curing:* The mold was left curing for a period of 48 h at room temperature of 25–27${}^{\circ }$. Small trapped air bubbles resulted in visible surface imperfections, but the measured diameter of such pores was less than 0.3 mm.

### 2.2. Image Registration Model

The influence of motion on iPPG signal extraction is best illustrated by Figure 2a. When an object is subjected to a motion vector, its relative position within a captured camera frame also changes. If a selected region of interest (ROI), used to spatially average pixel values within its boundaries, is expected to be at a location $X_{1}$-$Y_{1}$ in Frame A, then this ROI drifts to occupy a location $X_{2}$-$Y_{2}$ in Frame B (see Figure 2a). This effect has two implications: (1) each camera sensor pixel focuses on a different tissue area during object’s motion resulting in iPPG observation inconsistency; (2) the underlying structure of skin tissue is expected to be spatially heterogeneous, so the intensity of the backscattered light and the morphology of the iPPG signal varies as the object moves *w.r.t.* the sensor, resulting in abrupt signal noise. Therefore, there is a need for tracking to focus on the same tissue area when extracting iPPG signals from the captured frames.

Optical image registration is often utilized to accurately work out frame-to-frame drift and refocus the ROI so as to consistently overlay the same tissue area. These are typically divided into feature-based and intensity-based groups. The former group aims to establish a relationship between some distinct pixels (features) in the images, such as corners, lines, or color differences, while the latter group is based on finding and comparing intensity patterns using various correlation metrics. The complexity of the registration process and the choice of the appropriate method depends widely on the type of geometric deformation present in the target images, its amplitude and frequency, as well as the number of resolution levels involved. A combination of geometric transformation functions is then applied to these feature or intensity patterns, pair-wise, in order to establish a motion vector and refocus the ROI (Figure 2b).

#### 2.2.1. Feature-Based Registration

Feature-based registration relies on the fact that images tend to have distinctive elements (features) associated with a pixel cluster that is different to its immediate neighbors in color or intensity. High quality features allow localization of the correspondences within images regardless of any change in the illumination level, view point, or partial occlusion. Two fundamental building blocks required for a high accuracy image registration include a feature detector and a feature descriptor.

Feature detection is a step responsible for identifying local corners, sharp edges, and blobs by utilizing intensity variation and gradient approaches. Selected features should be distinctive and exhibit a significantly different pixel-value gradient compared to other neighboring pixels (Figure 2b). The corresponding features should also demonstrate a uniquely assigned location and remain locally invariant, which ensures accurate registration even if the viewing angle has changed.

A feature descriptor is a compact vector representation of a particular local neighborhood of pixels. This step establishes an accurate relationship between the target and reference images which remains stable in the presence of noise, image degradation, and changes in scale or orientation, while properly discriminating among other feature pairs. The choice of a descriptor is likely to be application-specific. Images containing high amounts of distortion benefit from computationally intensive local gradient-based descriptors, such as KAZE [20] and SURF (Speeded Up Robust Features) [21]. Binary descriptors like FREAK (Fast Retina Keypoint) [22] and BRISK (Binary Robust Invariant Scalable Keypoints) [23], which rely on pairs of local intensity differences encoded into a binary vector, are generally faster but less accurate than gradient-based descriptors.

To address a wide range of potential distortions due to the palm motion, the SURF feature detector and descriptor were selected (denoted as *Feature-based* in this paper). Palm skin surfaces, examined under visible light, do not generally exhibit many sharp corners or high contrast lines, making utilization of FREAK and FAST (Features from Accelerated Segment Test) [24] detectors less suitable. Therefore, the use of a feature descriptor aimed at detecting regions that differ in brightness or color compared to surrounding regions is justified at the cost of increased computational load.

#### 2.2.2. Intensity-Based Registration

Intensity-based registration algorithms align images based on their relative intensity patterns. A specific performance measure called a similarity metric is computed and iteratively optimized, thus producing the best possible alignment results for a given set of images and geometrical transformations (see Figure 3). This optimization problem may be expressed as
(1)$$\mu_{min}=\arg\min_{\mu }\Psi \left(I_{R},T^{\left(\mu \right)}I_{T}\right)$$
where $T^{\left(\mu \right)}$ is an intensity transformation, mapping the target image $I_{T}$ with transformation parameter vector μ, and Ψ is the similarity metric between the reference image $I_{R}$ and the transformed target, and is used with the goal of finding $\mu_{min}$, which minimizes the difference between the two. The most common similarity metrics include the sum of absolute differences [25] or the mean squared error. A simple implementation of Equation (Equation 1) to an image sequence $I_{n}$ compares a translation $T^{\left(\mu \right)}$ of the frame $I_{n+1}$, with displacement test vector μ=(α,β) pixels, to the preceding frame $I_{n}$ using a sum of absolute differences (SAD) similarity approach across the ROI of frame $I_{n}$. One obtains a “best fit” displacement vector as the translation that minimizes the SAD metric, giving an estimate of the motion vector as
(2)$$\mu_{min}=\left(\hat{\alpha },\hat{\beta }\right)=\arg\min_{\alpha ,\beta }\sum_{\left(j,k\right)\in ROI_{n}}\left|I_{n}\left(j,k\right)-I_{n+1}\left(j+\alpha ,k+\beta \right)\right|.$$

The minimization here is performed over some specified pixel range appropriate the type of motion involved. This particular form of Equation (Equation 1) is commonly seen, for example, in the compression of MPEG data prior to video transmission. In the general case, the optimizer block governs how the search for an ultimate image alignment should be conducted. The optimization problem is often tackled by an iterative strategy to achieve a minimal error between the registered images. Optimizers available in the literature include gradient descent [26,27], advanced adaptive stochastic gradient descent [28], simultaneous perturbation [29,30], Quasi-Newton [27,31], and nonlinear conjugate gradient [32,33] methods.

The biggest drawback encountered in many existing metrics is the effect of photometric distortions, such as changes in brightness and contrast [34]. Assumptions that pixel values are independent of each other and that the brightness stays constant, or exhibits spatially stationary changes, are only valid in specific cases, so the selected technique should be able to account for illumination changes. Two intensity-based registration methods are evaluated in this paper: one based on a mean square error metric and a regular step gradient descent optimizer (denoted by *Intensity 1*), together with a more advanced one based on the enhanced correlation coefficient (ECC) [35] (denoted by *Intensity 2*).

#### 2.2.3. Frame Registration

Motion was estimated between each pair of neighboring frames, i.e., $I_{n-1}$ and $I_{n}$, $I_{n}$ and $I_{n+1}$, $I_{n+1}$ and $I_{n+2}$, etc. This method is called *frame-by-frame* in this paper. The offset of $I_{n}$-th frame *w.r.t.* the first image is a cumulative sum of the $\left(I_{1},I_{2},I_{3},...I_{n-1},I_{n}\right)$ sequence.

A normalized cross-correlation (denoted by *Norm X-correlation or NCC*) was also used as a simple and computationally inexpensive registration method usually employed by researchers to be compared with more advanced motion compensation schemes. Intensity- and feature-based registration was performed at multiple resolution levels, where the first coarse alignment was obtained using down-scaled versions of the input images to remove the local optima and reduce computational load. The algorithm entered the next more detailed level once the optimizer converged to a suitable solution, down to a sub-pixel level for a more precise registration, using a linear interpolation scheme.

It should be noted that the image registration schemes can suffer from convergence issues, where the search for the optimal solution stops prematurely due to the presence of local optima. The usual sources of misconvergence include a step size being set at too large a value to minimize computational time, the number of resolution layers being too small to detect finer motion, the registered images not having enough overlap to establish corresponding feature pairs or intensity regions, and the algorithm not being able to detect features to accurately estimate the geometrical transformation. As a result, the incorrectly estimated pixel offset can lead to zero (no motion detected) or start oscillating (excessive motion and inability to track the object correctly).

### 2.3. Hardware Setup

This study consists of two parts: (a) a comparison of image registration methods using a silicone palm model and (b) an evaluation of the selected registration methods using live subjects.

Since the quality of iPPG depends on complex light-tissue interactions, the amplitude and morphology of a signal could be influenced and modulated by instability of the incident light, so the ability to control a light source in a reproducible manner is vital. The effects of variations in instrumentation equipment have not been fully evaluated in the literature, thus making iPPG system behavior less certain, and its potential applications less attractive to the practical community.

In order to create a stabilized light flux, an illuminator comprised of individual high-power LEDs was constructed based on the principles and results reported in previous work [36]. In short, three high-power light emitting diodes (LEDs) (LXML Series, Philips, Andover, MA, USA) emitting at 530 nm with a typical half-power bandwidth of 20 nm were mounted on an aluminum substrate of high thermal conductivity to divert excess heat generated by the semiconductors (see Figure 4a). When operating at full power the system was capable of delivering 470 lm of luminous flux. Current flow stability is achieved by designing a constant-current LED driver capable of varying the forward current using a 12-bit digital-analog converter (DAC) in order to achieve different levels of light intensity.

The selected LEDs were manufactured with a dome-shaped micro lens to provide a 120${}^{\circ }$ field of view (FOV). Placed at about 50 cm above the surface, the illuminator could produce a fairly uniform but dispersed flux, resulting in a dimmer ROI and additional stray reflection from nearby objects and surfaces. A 25${}^{\circ }$ FOV polycarbonate collimating lens (Carclo Optics, Aylesbury, UK) with 91% optical transmission was placed in front of the LED assembly to concentrate available flux on a smaller ROI. An additional 220-grit ground glass diffuser (Edmund Optics, Barrington, NJ, USA) with 75% transmission at 530 nm was placed in front of the collimating lens to diffuse light and even out non-uniformity caused by the circular lens.

A non-contact sensor comprised of an sCMOS monochrome camera (Orca Flash V2, Hamamatsu Co., Hamamatsu City, Japan) with an effective resolution of 2048 × 2048 pixels was coupled with a set of prime 50, 85, and 100 mm lenses (Planar T ZF-IR, Zeiss, Oberkochen, Germany) to provide several image magnification options and achieve higher system resolution. The camera was controlled from a workstation via a Camera Link interface and triggered by a control unit (CU) at 50 frames per second (fps). A contact PPG probe (cPPG) based on an integrated optical sensor (SFH 7050, OSRAM, Munich, Germany) sampled at 200 Hz was also placed on each subject’s finger to act as a reference point.

### 2.4. Experimental Protocol

The performance of image registration algorithms was evaluated on 10 subjects (aged 21–45) at the Photonics Engineering Research Group, Loughborough University, UK. These subjects belonged to Groups II and III of the Fitzpatrick skin classification system [19]. The experimental protocol was approved by the Ethics Committee at Loughborough University, UK, and all subjects signed a consent form prior to the experiment.

Each subject was asked to rest his/her palm on a support approximately 20 cm below heart level in a seated position. The support consisted of two thin and slightly oiled silicone sheets with low friction between them but with high friction between the skin surface and the sheet. The bottom sheet was placed inside a redesigned rectangular lid with 20 mm high borders around its perimeter, fixed to a table surface, so the palm was constricted to move by no more than ±15 mm in any direction (Figure 4b). Subjects were instructed to relax and move their palms freely without concentrating on the amplitude or frequency and were also asked to limit their hand supination and pronation. A contact PPG (cPPG) probe was strapped to the middle finger to act as a reference signal. The camera and lens assembly were positioned above the palm on an adjustable arm. The lens was focused and centered on the area of the highest LED illumination level verified by a real-time histogram on PC software. The image frames were captured during four separate sequences each lasting 12–15 s.

The goal of this study is to minimize small artificially induced motion similar to what a generic patient with tremors, respiratory-related movement, or involuntary muscular contraction can experience in a hospital environment during a body part scanning. Cases involving significant motion, such as in head or palm rotation or tilting, are unlikely to benefit significantly from the application of image registration, as different, previously unexposed, skin regions may be recorded by the camera.

### 2.5. Signal Processing

Figure 5 shows the framework utilized in this study. The video frames and the contact reference signal were processed offline using Matlab (Mathworks, USA) pipelines and algorithm implementations. Image scaling was performed on the raw frames to simulate the effects of system resolution, or an altered distance between the camera and an object, on the image registration and extracted iPPG signals.

For a fixed sensor-lens pairing, an increase in distance between an object and the optical system would result in a single pixel covering a wider area on the object’s surface, making the image less resolved and potentially causing smaller details to disappear (Figure 6a,b). The region covered by a single pixel may also be altered by changing the focal length of a lens (provided the focus is maintained); a shorter length and a wider viewing angle could result in a larger area covered by a unit pixel.

The resolution is normally defined as an ability of a given imaging system to reproduce individual object details (Figure 6). An increase in the pixel count, usually referred to as a higher pixel resolution, causes each sensor element to cover a smaller object area, provided all other factors remain constant. Consequently, finer palm surface details are easier to resolve and capture by individual pixels as the image resolution increases (Figure 6b,c). Our hypothesis here is that a higher system resolution, and a more detailed image, should facilitate better frame registration and track smaller motion distortions by providing better resolved anchor points for an algorithm to use.

To simulate a reduced system resolution, the frames were captured at the camera’s maximum allowed pixel resolution (2048×2048) with the lens set at a distance to cover the whole palm, resulting in 74 pixels/cm. The image was progressively rescaled offline from the original size down to a ×0.2 factor using linear interpolation methods.

#### 2.5.1. Signal Formation

All pixels within the selected ROI were spatially averaged, once successfully registered. This procedure was repeated for all frames in the data set resulting in a time-varying signal. The video sequences obtained by filming the silicone palm do not exhibit any periodic cardiac-induced variations, so its zero-mean temporal signal should ideally be zero. The video series capturing live tissue, however, include a slowly-varying *quasi*-DC signal, while the cardiac cycles could vary in their amplitudes. The effects of applying optional normalization methods, such as division by its lowpass-filtered component, polynomial fit, or normalization with a Gaussian distribution are discussed in Section 3.2.1.

#### 2.5.2. PPG Analysis

To evaluate the effect of motion compensation on signal composition, the extracted signal was analyzed in the time and frequency domains from five data sets. Separate methods were applied to the silicone palm and live tissue samples. The absolute noise floor was established using a signal derived from the silicone palm with no induced motion. In ideal conditions, the spatially averaged signal should be DC-stable, and any deviation indicating the presence of noise in either the optical imaging or the light source.

For live tissue signals, the evaluation criterion included noise component analysis in the frequency domain, adapted from [37]. Ideally, a PPG signal has a clear and distinct peak with sharp roll-offs around its fundamental pulse frequency, followed by a small number of harmonic components. These frequency components would be attributed to a clean PPG signal. The signal-to-noise ratio (SNR) was calculated to act as a quality metric, where the *signal* component was estimated from the spectral energy within ±0.1 Hz of the fundamental heart rate frequency identified and located by the reference contact (cPPG) signal, and the *noise* was estimated as the remaining spectral energy in the 0.5–7 Hz range.

A heart rate (HR) was extracted in the frequency domain by obtaining a fast Fourier transform (FFT) of a normalized PPG signal. The position of the frequency component with the highest amplitude was located, which corresponded to the heart rate with the conversion factor of 1 Hz = 60 BPM. cPPG was assumed to be less susceptible to motion artifacts and less noisy due to its firm contact with the fingertip, so it was selected as a ground truth signal for the current experiments.

## 3. Results

### 3.1. Experiments with Prosthetic Palm

#### 3.1.1. Influence of Region Size

The palm replica was filmed under green light illumination and rescaled with pseudo colors as illustrated in Figure 7a. The highest and lowest pixel intensities are represented by shades of red and green, respectively. Since there was no backscattered light modulated by the cardiac activity in this experiment, the registered pixel intensity was directly influenced by the specular reflection of the palm surface and any residual reflection from the nearby surfaces. The contribution of these reflections depends largely on the light uniformity within the ROI, the angle between the optical devices (the camera and the light source) and the palm surface, and the optical properties of the material used in lifecasting. The flat, top surface (the red region near the top-right corner of ROI 1 in Figure 7a) was orthogonal to the camera and the light source, with the specular reflection resulting in a significantly higher intensity compared to other regions. Pixel intensity non-uniformity due to the palm curvature is visible toward the edges of the palm (cyan and green shades). Therefore, a shallower light incident angle caused lower specular reflection, resulting in reduced pixel values observed by the camera sensor.

If the ROI is not subject to motion and the illumination source is *quasi*-stable between individual image exposure periods, then the extracted spatially averaged signal is also expected to be *quasi*-DC with potential small fluctuations attributed to the quantization noise and the light source instability. Therefore, image registration should result in zero offset in both the *X*- and *Y*-axis. Figure 7b,c confirm that motion predictions were close to zero values, but all except the NCC method showed oscillations with deviations of up to 0.25 pixels from the reference level. These results should be interpreted as a minimal noise floor for the given system and a particular image registration method, established by calculating the standard deviation ($\sigma_{\bar{I}}$) and the peak–peak amplitude of a spatially averaged signal extracted from a static ROI (Figure 7d). From this, one would expect that a live tissue iPPG signal will possess fluctuations of at least 0.16% its DC value, even if the ROI is stable. Assuming that iPPG peak–peak amplitude could be as low as 0.6% of its mean temporal signal when filmed under visible light [18,38], the noise floor could reach a quarter of the available signal headroom. As a result, it seems necessary to establish the value of this parameter as a figure of merit, perhaps using an inanimate object placed adjacent to the imaged body part, when reporting on the iPPG signal quality.

These various combinations of ROI sizes and locations were evaluated in order to establish the relationship between the motion intensity and the quality of image registration. Figure 8 depicts arbitrary motion sequences from two areas shown in Figure 7a, where image registration is applied (here, ROI 1 is 8.2 times larger than ROI 3). When plotted side by side, the difference in motion estimation is clearly visible. Within the larger region (ROI 1), all four registration methods were aligned in their prediction, which, for this particular trial, corresponds to up to 35 pixels in the X-direction and about 8 in the Y-direction, or 10% and 4% of the ROI’s width and height, respectively. Intensity-Based Method 1 and the feature-based estimation showed the closest correlation among all approaches, with Intensity-Based Method 2 deviating no more than 0.8 pixel from these two. A single level NCC could not resolve sub-pixel registration without additional interpolation, resulting in its predictions being accurate to the closest integer value. However, the absolute difference between the NCC and other methods was within ±1.2 pixels.

The smaller region (ROI 3) clearly demonstrates the inability of the feature-based method to converge reliably, shown in Figure 8b as an oscillating signal without a specific frequency and pattern. Although, for a smaller motion amplitude (5–10 pixels), the feature-based registration was accurate to within ±1.5 pixels compared to other techniques, the remaining part of the registered images included significant error in shift estimation. Poor results were potentially influenced by the lack of enough distinguishable features within the ROI required for the method to converge properly. Moreover, the palm’s geometrical transformation was a comparatively large portion of the height and width of ROI 3; the registered motion was as much as 31% and 11% of ROI’s vertical and horizontal dimensions, leaving fewer mutual areas between the reference and moving frames for the strong feature pairs to be established. NCC and intensity-based methods demonstrated good correlation in their motion estimations in both regions.

Cross-correlation was used to compare motion estimation results between ROI 1 and ROI 3, normalized to the 0–1 range (Table 1). NCC showed the highest correlation of 0.99, although this could be due to the motion offset being rounded to the nearest pixel in a single-level registration rather than a fractional value as seen in for the multi-level intensity- and feature-based algorithms, while the feature-based method showed the poorest agreement between ROI 1 and ROI 3.

Based on the data analyzed from multiple video sets, Intensity-Based Method 1 showed the highest stability between datasets. This algorithm was further evaluated by comparing spatially averaged signals before and after the motion registration was applied. Figure 9 demonstrates two examples of signal improvement with the *Intensity 1* method applied. The largest fluctuations due to motion were significantly suppressed, with peak–peak variability reduced by a factor of up to 4.5 (Figure 9a,b). Another evaluation metric included the standard deviation ($\sigma_{\bar{I}}$) of the extracted signal spatially averaged across the ROI, where a low deviation indicated less noise and better motion immunity. This parameter was reduced by almost 5 times for Example 1, with a less effective, but still significant, reduction in Example 2 of 1.9 times (Figure 9c,d).

An interesting observation, in situations where the inter-frame motion was relatively small (Figure 10), is that object motion along the *X*-axis after Frame 200 was in excess of ±1 pixel, followed by good correlation between the NCC and intensity-based approaches. However, object deviation along the *Y*-axis during the 200–600 frame period was usually predicted at the sub-pixel level, with the single-level NCC rounding its result down to zero if the estimated shift was below 0.5 pixel. Consequently, the cumulative prediction in this region showed no object motion for the NCC algorithm (dotted line in Figure 10b), while the multi-level intensity-based methods aligned perfectly and the feature-based approach incorrectly predicted motion in the opposite direction (shown by the negative values).

#### 3.1.2. Influence of Resolution and Distance of Source

The results for the prosthetic palm dataset are presented in Figure 11 using the *Intensity 1* scheme. Resolution was reduced from the original of 74 pixel/cm in steps of ×0.2 using linear interpolation. The simulation showed an almost linear relationship between the resolution and registered offset in pixels, from 26.44 pixels at ×1.0 to 5.27 pixels at ×0.2, as shown in Figure 11a. With each individual camera pixel capturing a larger area as the system resolution decreases, it is expected that the same motion amplitude would result in fewer registered pixel offsets, to the point where the entire object displacement cannot be effectively resolved. This expectation was largely confirmed by running image registration on the downscaled frames followed by extracting spatially averaged signals. The results of the comparison are illustrated in Figure 11b together with the original uncompensated signal. The inclusion of motion compensation was beneficial for all resolution simulations, reducing the peak–peak fluctuation and their standard deviation ($\sigma_{\bar{I}}$) as seen in Table 2. The steepest performance reduction was observed between ×0.6 and ×0.4, where the peak–peak fluctuation increased by 30.2%.

### 3.2. Experiments with Skin Tissue

The first-stage experiments, conducted with the silicone palm, clearly demonstrated the inability of the feature-based frame registration method to accurately and reliably predict object motion due to the limit of well-defined anchor points. Therefore, only the three remaining image registration methods were selected for further tests on skin tissue.

Figure 12 shows the effect of motion compensation via image registration on the extracted iPPG signal. It is clear that all three methods show well defined cardiac-related pulses, together with 4–5 high amplitude cycles caused by the induced motion in Frames 290–450, with an amplitude range of up to 35 pixels (4.7 mm) (see Figure 12a). NCC managed to reduce this motion effect in Frames 300–450, but a low frequency step is still present. The intensity-based algorithms provide a better response to the same excitation, with the average DC level not changing dramatically before and after the induced motion region. Figure 12b illustrates a signal segment containing two cardiac cycles. High-frequency noise is clear in all four signals, partially inherited from the raw data and partially due to the undertaken image registration and multi-level interpolation. The application of image registration altered neither the peak–peak amplitude of a cycle, nor its visible morphology in terms of better defined peaks, troughs, and dicrotic notches, confirming the findings from the silicone palm test phase.

Spectral analysis uncovers a frequency range where the original uncompensated signal had a double peak with an amplitude higher than that of the expected cardiac-related region around 1.2 Hz. In Figure 12c, the spectrum (*I*) should be considered as carrying a heart rate component at 1.16 Hz, as verified by the contact pulse sensor. Instead another spectrum (*II*) could be potentially mistaken for a *true* heart rate, since it also lies in the potential cardiac-related region of 1.5–2.2 Hz or 90–132 BPM. Application of motion compensation techniques are able to clean that ambiguous spectrum around 1.5–2.2 Hz and better define the spectral range of interest (*I*), while additionally reducing energy in the sub-0.5 Hz spectrum with these *Intensity 1–2* methods.

Passing the uncompensated iPPG waveform (same as in Figure 12a) through a 0.6–3 Hz 3rd-order filter revealed motion-induced in-band noise with its frequency close to that in the pre-motion range (0–5 s), which can be mistaken for the desired pulse rate. Therefore, additional pre-processing to suppress that undesirable source, such as the frame registration discussed here, is clearly beneficial (Figure 13).

#### 3.2.1. Effects of Signal Normalization

The hypothesis that signal normalization in the time domain might facilitate motion artifact suppression without an explicit image registration stage was next put under test. Local illumination non-uniformity, low frequency non-cardiac oscillations, and small variations in specular and backscattered light due to a change in the skin surface orientation (angle) *w.r.t.* the camera system have all been accounted for by applying *AC/DC* normalization [39,40,41]. Three commonly found methods were utilized in this study, namely moving average, moving median, and low-pass filtering (LPF). The first two approaches were constructed with a moving window of 75 samples with a single frame step, while the LPF used 7th order Butterworth filter with 0.5 Hz cutoff. Original signals were padded at the beginning and end to allow filter buffers to initialize properly without edge discontinuities. Once *quasi*-DC components were obtained, the *AC/DC* normalization was performed as $iPPG_{norm}=iPPG_{raw}/iPPG_{DC}-1$.

Figure 14 summarizes the outcome of applying normalization to the motion-uncompensated signal. The improved results after eliminating a sharp low-frequency step between Frames 300 and 450 suggest partial success of this additional stage, as the cleaned signals have less peak–peak discrepancy between the true PPG cardiac cycles and motion-induced oscillations in the middle of the dataset (seen in the time domain in Figure 14a–c). However, spectral analysis reveals that the ambiguous double-peak previously seen in Figure 12c is smoothed out and moves closer to the assumed cardiac-pulse band. As can be seen in Figure 14e,f, the peak on the spectrum plot occurred at 1.68, 1.68, and 1.79 Hz, respectively, for the three normalized iPPG signals, while the actual heart rate was recorded and verified as 1.16 Hz by the reference contact cPPG probe. Therefore, uncompensated, but normalized, iPPG signals can still be harmed by in-band motion artifacts that can mask the cardiac pulse-related spectral peak and replace it with ambiguous values related to induced motion of other signal fluctuations around the expected cardiac-pulse frequency.

The signal-to-noise ratio metric described in Section 2.5.2, and aggregated for 10 subjects, is shown in Table 3. Although the application of image registration is not able to replicate the performance of the contact probe, it is able to boost the SNR value up to 8.34 from the original unadjusted 4.75. The results obtained from the normalization stage are clearly misleading since the energy around the fundamental cardiac pulse frequency included spectral content that belongs to motion-induced rather than cardiac activity. The frame registration process improved the SNR by 5.8% via the Intensity 2 method but was insignificant via Intensity 1. As expected, the NCC algorithm suffers from sub-pixel rounding in the frame-by-frame approach and demonstrates a 4.6% reduction in performance compared to a single-frame reference registration.

## 4. Discussion

In this study, the initial noise floor assessment was performed on a silicone cast of a human palm. The inclusion of this stage can be avoided by applying image registration techniques directly into a live tissue dataset. However, the initial algorithm and optical setup assessment shows clear benefits. Firstly, this approach helps to establish a performance baseline for each image registration algorithm, since the backscattered light and stray reflections do not contain intensity fluctuations related to blood volume variations during a cardiac or respiratory cycle. An iPPG signal extracted from live tissue, in contrast to a static cast, could contain a complex combination of a low-amplitude oscillating component due to the propagating cardiac pulse-wave, some noise inherited from the instability and transients in the optical system, as well as any noise residue left from or introduced by image stabilization. Our previous research concluded that even a very stable tissue sample can result in an iPPG waveform effectively buried in wide-spectrum noise. In a tissue sample, subjected to moderate motion, the iPPG signal quality is *a priori* unknown, and one can only guess whether its source is due to the initially poor and noisy iPPG signal or a badly tuned image registration process. Secondly, although live skin opto-physiological properties are not maintained when targeting a silicone mold, the prosthesis inherited the shape, surface roughness, and inhomogeneity of real skin. Therefore, the effects of palm motion *w.r.t.* the camera system can be modeled effectively, regardless of the dye pigment or the wavelength of the illuminator used.

An empirically established system resolution of 74 pixels/cm of tissue surface, projected into the camera sensor, is suggested as being sufficient to detect and rectify small amplitude motion, similar to those that might be experienced in a hospital environment by a generic patient with tremor, respiratory-related movement, or involuntary muscular contraction, during a body part scanning. Decreasing the system resolution from 74 to 14.8 pixels/cm (×0.2 of the full resolution) affects the extracted signals in a nonlinear fashion, with peak–peak fluctuations and pixel standard deviation $\sigma_{\bar{I}}$ rising from 0.42 to 0.85 and from 0.09 to 0.22, respectively (Table 2). Although the requirements of the optimal system resolution depend on the expected motion amplitude, the observed benefit of a higher resolution would be suitable for those applications demanding the lowest possible noise within the extracted signals, as well as those featuring finer object details, such as iPPG perfusion mapping. The gain obtained with such high resolution, however, comes at the cost of an increased computational load at the image registration stage.

The speed and accuracy of intensity-based registration depends on a number of parameters, including the number of registration levels and the choice of optimizer, the similarity metric, and the frame interpolation method. Generally only a small number of search steps is required before reliable convergence is achieved, and additional registration enhancement can be expected as parameters are tuned further. The choice of image interpolation scheme may also influence the amount of noise, and consequently the deviation from an ideal result. The number of image registration levels should be estimated from the minimal motion amplitude and the available system resolution. Observing that sub-pixel leveling involves further interpolation, and potentially the creation of additional noise, we recommend that a minimal sub-pixel registration leveling is used together with a higher system resolution optical setup, with a single camera pixel covering an area equal to (or smaller than) the average frame-to-frame object displacement.

Depending on the image size, and the complexity of object’s geometrical transformation during the course of the video sequence, the computational requirement for the whole-frame global registration may be too great for real-time motion stabilization and data analysis. Consequently, it may only be possible for registration of a smaller image segment, or region of interest (ROI), during the extraction of vital signs. The size of such an ROI plays an important role in determining the quality of motion compensation and the final extracted iPPG signal. Following an iterative search, an ROI of roughly 1.5×1.0 cm, measured on the palm surface, is the smallest recommended area compatible with achieving stable and repeatable results from the motion compensation algorithm. It is important to observe that these findings were obtained on human palm surfaces with relatively high surface homogeneity, and when viewed under green light; other body areas, such as the face, might render different results due to more well-defined local features, and skin imperfections, capable of acting as control points for better frame tracking.

The ultimate goal of an iPPG system, including any signal-enhancement algorithms, such as image stabilization, is its ability to run and display results in real-time, which will accelerate its acceptance and wide uptake in real clinical applications. Matlab software, used here for signal analysis, can have significant overhead in algorithm implementation depending on the underlying code and optimization paths selected by its developers, which limits our ability for online processing. The major performance bottleneck identified is the processing of a single frame pair at a time, taking on an average of 2.5–3 s on a generic consumer-grade 2.6 GHz processor. For real-time or near real-time analysis, the computational requirement is stringent and is around 10–30 ms per image pair depending on the frame rate. One workaround is to use parallel distributed computation, commonly found in graphics accelerators (GPUs). This approach will not only allow processing of 30–100 image pairs per second in parallel (provided ~1 s of video frames has been already buffered), but also the assignment of multiple cores to the same frame pair to achieve faster search for either control points or intensity regions. As similar problems are encountered in MPEG motion estimation, a number of fast optimization algorithms have been developed [42] to evaluate the matching necessary in Equation (Equation 2), and it is very likely that these can be adjusted for the present context. Finally, the utilization of application-specific electronic chips (ASICs) can yield substantial performance improvement by developing job-specific hardware and firmware without unnecessary overhead found in general-purpose solutions. In general, ASIC systems are only developed once a particular algorithm or method has been fully developed, has been tuned, and is ready for field deployment, but a number of MPEG ICs running SAD and other optimization algorithms have already been designed for both ASIC and FPGA platforms (see [42] and references therein).

Signal normalization by the temporal *quasi*-DC value is considered an alternative to optical image registration. Although such normalization mainly eliminates low-frequency distortion, it does not demonstrate substantial success in removing in-band distortion (Figure 14). Image registration followed by a normalization stage is proposed in order to suppress iPPG signal noise due to fluctuations in light uniformity, which is not compensated by frame registration.

A known restriction in this study is its limited ability to track the hand during its supination or pronation. Significant geometrical transformation, such as tilting, is not likely to benefit significantly from the application of image registration since new, previously unexposed, skin regions may be exposed to the imaging system. The influence of the frequency of the induced motion on the estimated object transformation has also not been evaluated during this project, and further research into this topic is advisable.

## 5. Conclusions

Due to the variability in optical system setups, a straightforward quantitative cross-comparison of motion compensation methods, applied to iPPG signals, remains difficult. The noise floor of an iPPG signal extracted from a static ROI may contain at least 0.16% of its DC value and reach around a quarter of the available signal headroom, ultimately masking some of the desired cardiac-related fluctuations. It seems sensible in further research in this area to also establish such a noise floor level using an inanimate object placed adjacent to the imaged body part, and we would encourage groups to quote such values, together with the *AC/DC* ratio, when reporting iPPG signal quality.

This study has lead to a number of recommendations in this area, including for the minimum system resolution and the ROI size, against which it will be useful to benchmark future work. The results also show that the application of frame registration reduces motion-induced in-band fluctuations, which might otherwise be considered as being cardiac-related, and it also outperforms the simple method of signal normalization which has previously been used to account for small non-cardiac oscillations. The study also finds that intensity-based methods are particularly effective in suppressing small motion of the human palm up to ±15 mm when illuminated by green light. Areas as small as 1.5 cm${}^{2}$ can be effectively stabilized with sub-pixel accuracy, thus enabling high-detail local PPG mapping.

Although the conclusions reached might not be common to all circumstances (including those of particular measurement site, or light wavelength used for illumination, or a particular motion frequency), the study suggests that integration of an intensity-based frame registration into an iPPG framework can facilitate filtering of both low- and high-frequency noise, while also significantly improving the cardiac-related spectrum of extracted iPPG signals.

## Figures and Tables

**Figure 1 sensors-18-04340-f001:**
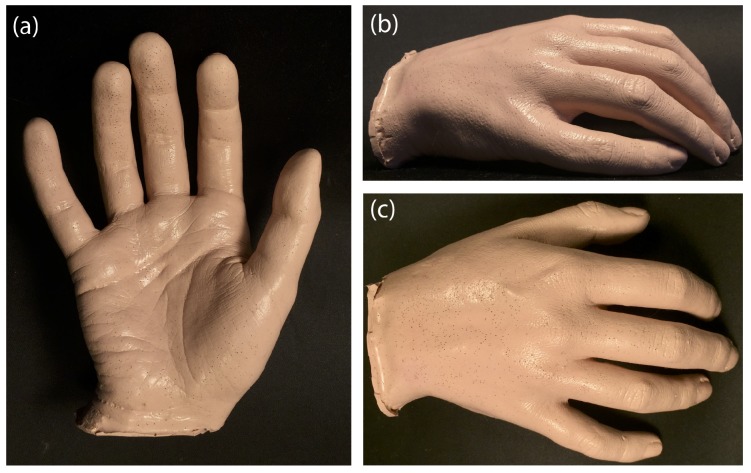
Three-dimensional copy of a human palm (female) with 1:1 scale reproduction, as seen from different angles (**a**–**c**).

**Figure 2 sensors-18-04340-f002:**
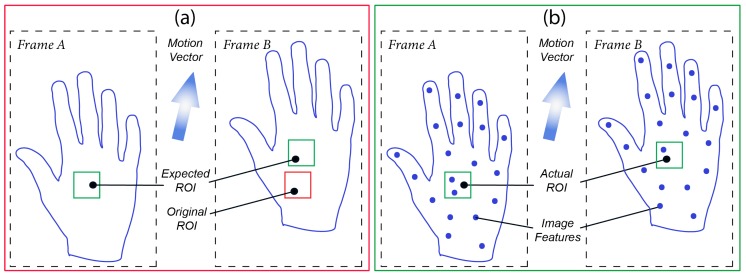
(**a**) Without image registration. The motion vector of the palm *w.r.t.* the camera sensor shifts the ROI away from its expected location in *Frame B*. (**b**) With image registration. A motion vector was computed and the shift was compensated by tracking the ROI to its actual position in *Frame B*, even though the object’s coordinates have changed in comparison to *Frame A*. Image features, such as hair follicles, deep wrinkles, birthmarks, or skin imperfections may be used to facilitate motion registration.

**Figure 3 sensors-18-04340-f003:**
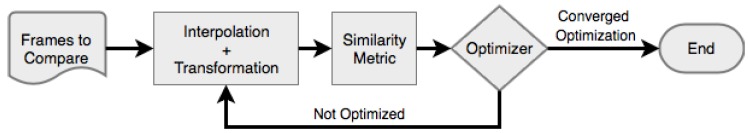
Iterative image registration process.

**Figure 4 sensors-18-04340-f004:**
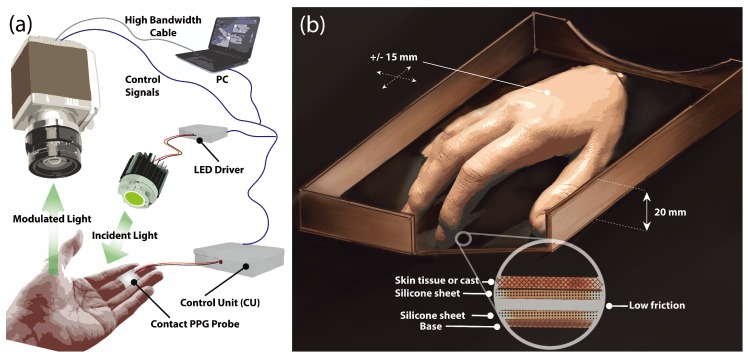
(**a**) Hardware setup for data acquisition. The control module (CU) is responsible for activating LEDs and triggering camera frame exposure using control signals. The intensity of LED assembly can be tuned via the CU. Parameters are set and monitored on a PC using a custom graphical interface. (**b**) The experimental setup showing the composition of the silicone sheets allowing the palm to slide freely in any direction within ±15 mm.

**Figure 5 sensors-18-04340-f005:**
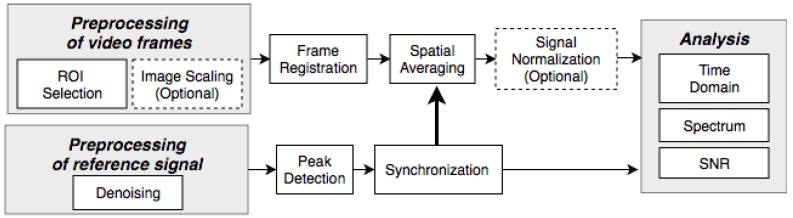
Signal processing framework used in this study. The contact reference signal was denoised by a 7 Hz low-pass filter.

**Figure 6 sensors-18-04340-f006:**
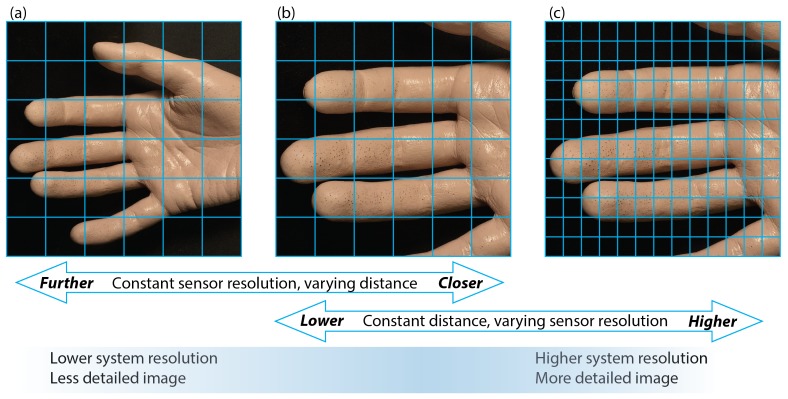
The effect of system resolution on the image detail level. Each square on the grid represents an individual pixel. (**a**,**b**) Varying distance or field of view, sensor resolution is left unchanged. (**b**,**c**) Varying sensor resolution for a fixed optical setup.

**Figure 7 sensors-18-04340-f007:**
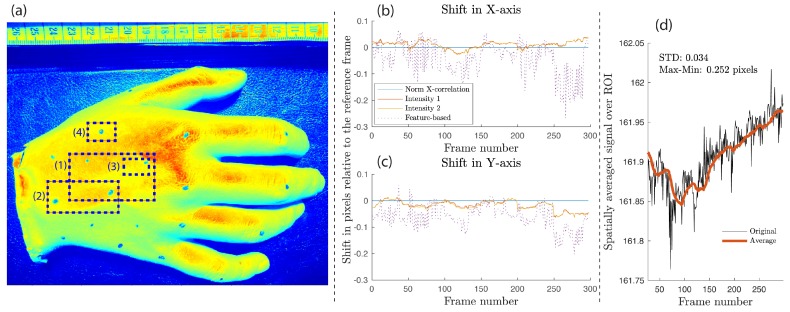
(**a**) Prosthetic palm replica with pixel intensity plotted in pseudo colors for better contrast. Regions of interest (ROIs 1–4) of various sizes were selected. Additional markers were added on the surface to test performance of feature-based image registration methods. Resolution was set at 74 pixels/cm. (**b**,**c**) Image shift along *X*-axis and *Y*-axis of a static silicone palm (ROI 1) should be zero, but sub-pixel variations for the intensity- and feature-based methods are clearly visible. (**d**) A spatially averaged signal extracted from a static ROI 3, with no registration applied, shows a deviation from its DC level, most likely due to the quantization noise and light source intensity variation.

**Figure 8 sensors-18-04340-f008:**
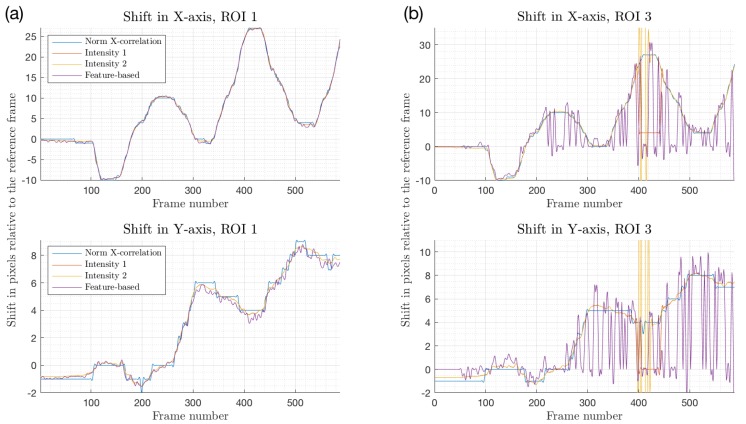
Motion registration results for *X* and *Y* axes when applied to ROI 1 (**a**) and ROI 3 (**b**). The ROI size was 341×188 pixels (4.6×2.54 cm) and 110×71 pixels (1.49×0.96 cm), respectively. All images in the dataset were (arbitrarily) registered *w.r.t.* frame #100. Negative values denote motion in the direction opposite to the positive predictions.

**Figure 9 sensors-18-04340-f009:**
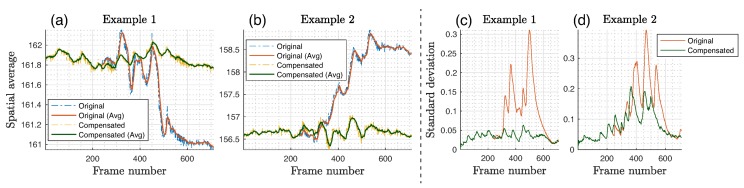
(**a**,**b**) Original (uncompensated) spatially averaged signals extracted from the silicone palm plotted with a motion compensated using Intensity-Based Method 1 to assess the effect of image registration in the time domain. Signals were passed through a simple 25-tap moving average filter (for illustration purposes) to suppress noise and show signal contours better. (**c**,**d**) Standard deviation ($\sigma_{\bar{I}}$) of the spatially averaged signal from an ROI was used as a metric to assess motion artifact reduction.

**Figure 10 sensors-18-04340-f010:**
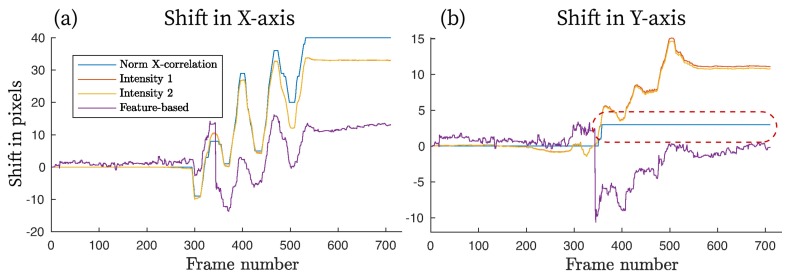
Effect of small inter-frame motion on image registration algorithm predictions. Notice a single-level NCC algorithm estimated no motion after Frame #350 caused by rounding the sub-pixel variations to zero (red dashed line).

**Figure 11 sensors-18-04340-f011:**
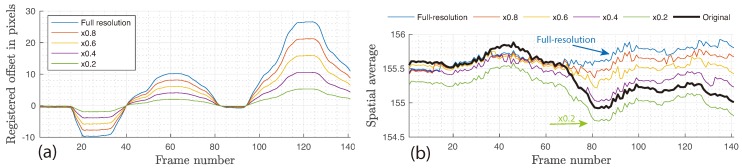
(**a**) Effect of system resolution on the image registration results along the *X*-axis and (**b**) on the spatially averaged signal extracted from ROI using the *Intensity 1* scheme. Both the width and height were progressively reduced from a full-scale ×1.0 down to ×0.2.

**Figure 12 sensors-18-04340-f012:**
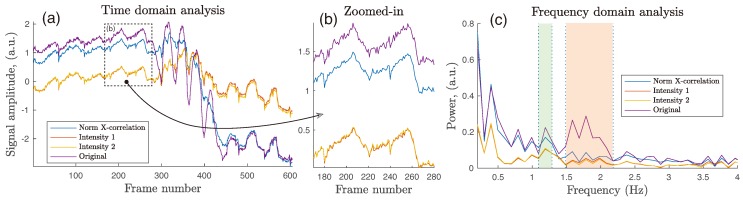
(**a**) Extracted iPPG signals with and without motion compensation applied. Signals were zero-centered by subtracting their temporal average. (**b**) Zoomed-in segment with vertical lines marking boundaries of cardiac cycles. (**c**) Frequency content of original and compensated iPPG signals. The green spectrum (*I*) is the expected cardiac-related region according to the reference contact probe cPPG, and the red spectrum (*II*) contains PPG harmonics and noise.

**Figure 13 sensors-18-04340-f013:**
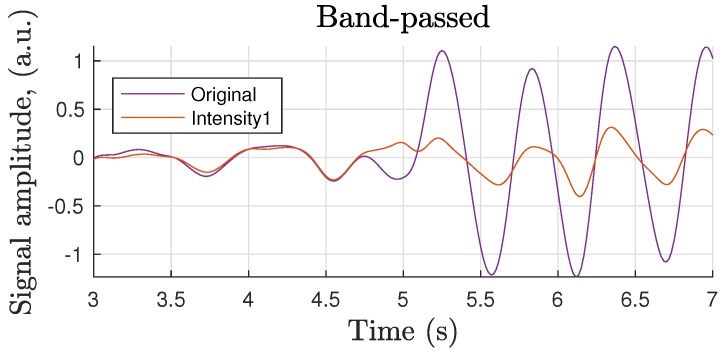
0.6–3 Hz band-pass filtered segment showing in-band motion-induced noise suppression by image registration.

**Figure 14 sensors-18-04340-f014:**
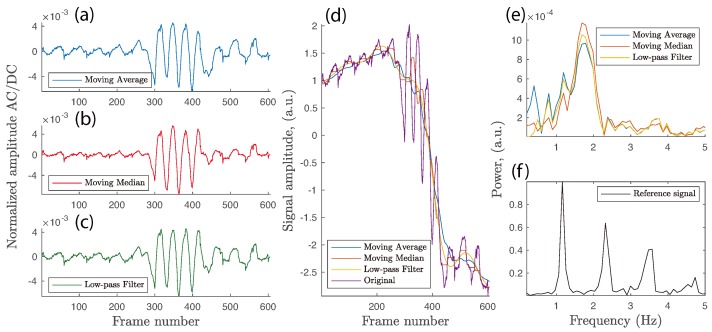
(**a**–**c**) Original uncompensated signal normalized using three methods. (**d**) *Quasi*-DC signals plotted against the original iPPG. (**e**,**f**) Spectral analysis of normalized uncompensated iPPG and contact reference cPPG signals, respectively.

**Table 1 sensors-18-04340-t001:** Cross-correlation between image registration algorithms for ROI 1 and ROI 3 in the *X*-axis and the *Y*-axis.

	NCC	Intensity 1	Intensity 2	Feature-Based
*X*-axis	0.99	0.84	0.71	0.67
*Y*-axis	0.99	0.96	0.73	0.71

**Table 2 sensors-18-04340-t002:** Results of peak–peak fluctuation and standard deviation before and after image registration obtained by simulating a range of system resolutions (pixels/cm of skin) for a prosthetic silicone palm.

	Full Resolution	×0.8	×0.6	×0.4	×0.2	Original Uncompensated
System~Resolution (pixels/cm)	74	59.2	44.4	29.6	14.8	74
Peak–peak~Fluctuation (pixels)	0.42	0.49	0.53	0.69	0.85	0.96
STD ($\sigma_{\bar{I}}$)	0.09	0.09	0.12	0.16	0.22	0.27

**Table 3 sensors-18-04340-t003:** Derived SNR metric for different motion compensation methods.

cPPG	Original	NCC	Intensity 1	Intensity 2	Moving Average	Moving Median	LPF
34.29	4.75	5.27	7.85	8.34	10.51	6.83	10.57

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
