# Peer review of "Frame Registration for Motion Compensation in Imaging Photoplethysmography"

_sensors, 2018, doi:10.3390/s18124340_

Round 1

Reviewer 1 Report

attached.

Reviewer 2 Report

Please, define the Equation more clearly than below

162 where T (m) is the intensity transformation, mapping the target image IT with transformation parameter
163 vector m, and Y is the similarity metric between the reference image IR and the transformed target; and
164 is used with the goal of finding mmin which minimizes the difference between the two.

Author Response

please see the attachment as requested.

Round 2

Reviewer 2 Report

According to the text "The first stage experiments, conducted with the silicone palm, clearly demonstrated the inability of the feature-based frame registration method to accurately and reliably predict object motion due to the limit of well-defined anchor points" so why this was done and reported here?

Author Response

See the response as attached.
